# Phosphonopeptides Revisited, in an Era of Increasing Antimicrobial Resistance

**DOI:** 10.3390/molecules25061445

**Published:** 2020-03-23

**Authors:** Emma C.L. Marrs, Linda Varadi, Alexandre F. Bedernjak, Kathryn M. Day, Mark Gray, Amanda L. Jones, Stephen P. Cummings, Rosaleen J. Anderson, John D. Perry

**Affiliations:** 1Department of Microbiology, Freeman Hospital, Newcastle upon Tyne NE7 7DN, UKnotthisearth@hotmail.com (K.M.D.); 2Department of Applied Sciences, Northumbria University, Newcastle upon Tyne NE1 8ST, UK; amanda.l.jones@northumbria.ac.uk (A.L.J.); S.Cummings@tees.ac.uk (S.P.C.); 3Sunderland Pharmacy School, University of Sunderland, Sunderland SR1 3SD, UK; linda.varadi@rmit.edu.au (L.V.); abedernjak@reviral.co.uk (A.F.B.); mark.gray@sunderland.ac.uk (M.G.); roz.anderson@sunderland.ac.uk (R.J.A.)

**Keywords:** phosphonopeptides, alafosfalin, carbapenemase, antimicrobial resistance, glycopeptide-resistant enterococci, MRSA

## Abstract

Given the increase in resistance to antibacterial agents, there is an urgent need for the development of new agents with novel modes of action. As an interim solution, it is also prudent to reinvestigate old or abandoned antibacterial compounds to assess their efficacy in the context of widespread resistance to conventional agents. In the 1970s, much work was performed on the development of peptide mimetics, exemplified by the phosphonopeptide, alafosfalin. We investigated the activity of alafosfalin, di-alanyl fosfalin and β-chloro-L-alanyl-β-chloro-L-alanine against 297 bacterial isolates, including carbapenemase-producing Enterobacterales (CPE) (*n* = 128), methicillin-resistant *Staphylococcus aureus* (MRSA) (*n* = 37) and glycopeptide-resistant enterococci (GRE) (*n* = 43). The interaction of alafosfalin with meropenem was also examined against 20 isolates of CPE. The MIC_50_ and MIC_90_ of alafosfalin for CPE were 1 mg/L and 4 mg/L, respectively and alafosfalin acted synergistically when combined with meropenem against 16 of 20 isolates of CPE. Di-alanyl fosfalin showed potent activity against glycopeptide-resistant isolates of *Enterococcus faecalis* (MIC_90_; 0.5 mg/L) and *Enterococcus faecium* (MIC_90_; 2 mg/L). Alafosfalin was only moderately active against MRSA (MIC_90_; 8 mg/L), whereas β-chloro-L-alanyl-β-chloro-L-alanine was slightly more active (MIC_90_; 4 mg/L). This study shows that phosphonopeptides, including alafosfalin, may have a therapeutic role to play in an era of increasing antibacterial resistance.

## 1. Introduction

The increasing resistance of pathogenic bacteria to antimicrobial agents is a substantial challenge to microbiologists and the medical community in general. There is a worrying lack of new antimicrobials in the pharmaceutical ‘pipeline’ and the lack of options for treatment of multi-resistant Gram-negative bacteria, such as those producing carbapenemases, has been recognised for some years as a particular concern [1,2,3]. A number of proposals have been made to address this issue, including the introduction of new incentives for the pharmaceutical industry to return their attention to antibacterial drug development. As an interim solution, it may be expedient to revisit old or abandoned antibacterial compounds to re-evaluate their efficacy against infections for which there are few remaining treatment options [4]. The increasing role of agents such as colistin and fosfomycin as treatment options for carbapenemase-producing Enterobacterales lends credence to this approach [5,6]. 

In the late 1970s, there was much interest in the use of peptide ‘mimetics’ as antibacterial agents. These typically consisted of an antibacterial compound covalently linked to one or more amino acids to facilitate uptake via the microbial peptide transport system. Intracellular hydrolytic cleavage (via aminopeptidase activity) would then release the antibacterial ‘warhead’ to interact with its target. Probably the best known agents of this type were synthesised by Roche and exemplified by alafosfalin (L-alanyl-L-1-aminoethylphosphonic acid). Bacterial uptake of alafosfalin is accomplished via LL-dipeptide permeases and subsequent hydrolysis yields fosfalin (L-1-aminoethylphosphonic acid) which binds with alanine racemase; thus preventing synthesis of D-alanine, an essential ingredient for peptidoglycan biosynthesis [7]. Alafosfalin was shown to have a broad spectrum of antibacterial activity [8] against certain Gram-positive aerobes (*Staphylococcus aureus*, *Enterococcus faecalis*, but not Group A streptococci and *Streptococcus pneumoniae*), anaerobic bacteria (*Bacteroides* spp. and *Clostridium perfringens,* but not *Clostridium difficile*) and many species of Enterobacterales (but not *Pseudomonas* or *Acinetobacter*). Against Gram-negative bacteria, alafosfalin is bactericidal, causing rapid lysis of susceptible strains [9].

Other synthetic antibacterials developed around the same time exploited a similar route of delivery and mode of action. For example, Cheung et al. explored the antibacterial activity of a range of halogenated dipeptides exemplified by L-β-chloroalanyl-L-β-chloroalanine. The β-chloroalanine released by hydrolysis interacts with a number of enzyme systems including alanine racemase and also shows broad spectrum antibacterial activity [10]. 

In this study, we report the evaluation of three peptide mimetics: alafosfalin, di-alanyl fosfalin and β-chloro-L-alanyl-β-chloro-L-alanine (compounds **A**–**C**; see Figure 1 for structures, full chemical names and abbreviations). These compounds were evaluated for antibacterial activity with a collection of 297 bacteria that included a predominance of multi-drug resistant strains, including carbapenemase-producing Enterobacterales (*n* = 128), methicillin-resistant *Staphylococcus aureus* (*n* = 37) and glycopeptide-resistant enterococci (*n* = 43). Fosfomycin, a naturally occurring antibiotic also containing a phosphonic acid group (**D**), was included for comparison.

## 2. Results

Table 1 and Table 2 shows the minimum inhibitory concentrations (MICs) for the four antimicrobials against the major groups of bacteria tested. Alafosfalin showed good activity against most isolates of Enterobacterales, although different species showed different degrees of susceptibility. Strong activity was shown against 53 isolates of *Escherichia coli*, of which 35 isolates (66%) were carbapenemase producers. The MIC_90_ for *E. coli* was 0.25 mg/L and all isolates were inhibited by 2 mg/L. The activity of alafosfalin was around fourfold higher than that of fosfomycin. *Klebsiella pneumoniae* was less susceptible to all of the test compounds when compared with *E. coli*, although many isolates showed relatively low MICs, e.g., 87% of *K. pneumoniae* isolates were inhibited by 8 mg/L alafosfalin. All isolates of *Enterobacter cloacae* (*n* = 27) were inhibited by 4 mg/L alafosfalin, which was typically 16-fold more active than fosfomycin against this species. Other species of Enterobacterales are not summarized in Table 1 as less than 10 isolates were tested. For *Citrobacter* species (*n* = 9), *Klebsiella aerogenes* (*n* = 1), *Kluyvera* sp. (*n* = 1), and *Serratia marcescens* (*n* = 1) all isolates were susceptible to ≤4 mg/L alafosfalin. One of five isolates of *Klebsiella oxytoca* required a MIC of >8 mg/L alafosfalin. Eight isolates of *Proteus mirabilis* and two isolates of *Providencia rettgeri* required MICs of ≥8 mg/L for all agents tested (including fosfomycin). Three isolates of *Salmonella* species required MICs of ≥8 mg/L for alafosfalin, whereas fosfomycin was more active against *Salmonella* (MICs: 0.125–0.25 mg/L).

The results of chequerboard synergy testing are shown in Table 3. For 16 of 20 Enterobacterales producing the five most common carbapenemases, synergy was observed between alafosfalin and meropenem as determined by a FICI ≤ 0.5.

Compared to alafosfalin, β-Cl-Ala-β-Cl-Ala showed relatively poor activity against Gram-negative bacteria but it showed the highest activity against *Staphylococcus aureus*; however, there was little difference overall between the three peptide-based antimicrobials. Ninety percent of methicillin-resistant *S. aureus* (MRSA) isolates were inhibited by 8 mg/L alafosfalin. Against enterococci, the most notable observation was the high activity of di-alanyl fosfalin, for which the MICs were (on average) 16-fold lower than those of alafosfalin and in some cases 256-fold lower (Table 2). Of 34 isolates of *Enterococcus faecium* (including 31 glycopeptide-resistant isolates), all were inhibited by 32 mg/L alafosfalin or 4 mg/L Di-alanyl fosfalin.

## 3. Discussion

The re-evaluation of previously abandoned antibacterial agents is a credible interim solution to the problem of dwindling treatment options created by increasing bacterial resistance. To our knowledge, alafosfalin was never licensed for clinical use and consequently there are no approved breakpoints to define susceptibility, but there are a number of published reports on the experimental use on alafosfalin in animals and volunteers [9,11,12,13,14]. Treatment of infections has been demonstrated in animal models, such as the mouse septicaemia model employed by Allen et al. that showed successful treatment of *E. coli*, *K. pneumoniae* and *E. faecalis* using alafosfalin [9]. In a study of human volunteers, Allen and Lees reported that oral doses ranging from 50–2500 mg of alafosfalin were well absorbed and well tolerated [12]. However, after oral administration of the drug, some hydrolysis prior to absorption resulted in a bioavailability of approximately 50%. In a further study with volunteers, Welling et al. confirmed that alafosfalin is rapidly absorbed with peak serum concentrations of 4–9 mg/L appearing within 1 h of a 500 mg oral dose [14]. It is also rapidly eliminated by the kidneys with a serum half-life of 1 h or less. They also reported that bioavailability following oral dosing could be significantly improved if the dose was given with milk rather than water. More recent work has shown that alafosfalin is actively transported across the intestinal epithelium due to its high affinity for H+ / peptide symporters, PEPT1 and PEPT2 [15]. In animal models, Allen et al. have reported that alafosfalin shows no significant binding to serum proteins and is distributed in most tissues except for liver and brain. In human volunteers, the same group reported that administration of 200 mg (intramuscular) or 500 mg (oral) doses consistently resulted in peak plasma levels of 5–10 mg/L with mean urine concentrations in excess of 50 mg/L for the first 6 h after dosing [11]. These concentrations were maintained on chronic administration but urine levels were adversely affected in patients with impaired renal function. The rate of absorption and elimination of alafosfalin was found to be similar for published data on β-lactam antibiotics [11] and synergy with β-lactams such as cephalexin and mecillinam, as well as other cell-wall inhibitors (e.g., D-cycloserine), has been well documented due to a ‘multi-blockade’ of cell wall biosynthesis [9,13,16,17]. 

Alafosfalin has a number of advantageous properties as an antimicrobial including a broad spectrum of activity, low toxicity, good absorption, low protein binding, a bactericidal mode of action, and a target site that is distinct from that of other licensed antimicrobials. It also has a number of limitations including its tendency to be metabolized (thereby reducing bioavailability), a reduced activity at alkaline pH, a relatively high ‘inoculum-effect’, and, according to some reports, the relative ease with which bacteria can acquire resistance [8,18]. Despite this latter claim, in a comprehensive review Ringrose concluded that spontaneous mutants with resistance to alafosfalin arise at a relatively low frequency of 10^−8^ to 10^−9^ due to the need for mutations in multiple permeases [19]. Given the limitations, it is understandable that alafosfalin was not actively pursued in an era when a wide range of alternative antimicrobial agents were available for most infections.

However, some of these limitations are shared by other antimicrobials that are relied upon for routine treatment. The ease with which bacteria can develop resistance to alafosfalin can be reduced by combining it with other established antimicrobials as demonstrated previously [13]. This study has shown that alafosfalin in combination with meropenem has a synergistic interaction against many highly resistant isolates of carbapenemase-producing Enterobacterales. We have shown that alafosfalin (and other peptide-based antimicrobials) retains strong activity against some bacterial isolates that are now difficult to treat using conventional antimicrobials. Given the large amount of data already available on the pharmacokinetics and safety of alafosfalin, it may be worth revisiting this agent as a potential treatment for infections caused by isolates that are otherwise difficult to treat. 

Phosphonopeptides such as alafosfalin have a unique target site that is not shared by any other antimicrobial agent in current use, and further research is warranted to ensure that this does not remain unexploited. As we have shown in this study, other peptide mimetics, such as di-alanyl fosfalin, may show substantial antimicrobial activity against *E. coli* and particularly enterococci. However, at present there is insufficient information on the pharmacokinetics and toxicity of other compounds to speculate further on their therapeutic potential. Atherton et al. have recommended other phosphonopeptides with improved potency and broader antibacterial spectrum [20] and these are also worthy of further investigation in the era of widespread antimicrobial resistance to conventional antimicrobials.

## 4. Materials and Methods 

### 4.1. Antibacterial Agents and Media

Fosfomycin, alafosfalin, glucose-6-phosphate and all ingredients of the antagonist-free agar were purchased from the Sigma Chemical Company, Poole, UK. IsoSensitest agar was purchased from Oxoid, Basingstoke, UK. The synthesis and characterisation of the two other antimicrobial agents is described in Appendix A. 

### 4.2. Bacterial Isolates

Enterobacterales (*n* = 197) were obtained from diverse international sources and all possessed β-lactamases that had been defined at a molecular level by reference laboratories and/or recognized experts in the field. These included *Citrobacter freundii* (*n* = 5)*,* other *Citrobacter* species (*n* = 4), *Klebsiella aerogenes* (*n* = 1), *Enterobacter cloacae* (*n* = 27), *Escherichia coli* (*n* = 53), *Klebsiella oxytoca* (*n* = 5), *Klebsiella pneumoniae* (*n* = 87), *Kluyvera species* (*n* = 1), *Proteus mirabilis* (*n* = 8), *Providencia rettgeri* (*n* = 2), *Salmonella* species (*n* = 3), and *Serratia marcescens* (*n* = 1). Of these 197, there were 128 (65%) carbapenemase producers including isolates with NDM-1 (New Delhi Metallo β-lactamase; *n* = 87), IMP (imipenemase; *n* = 9), KPC (*Klebsiella pneumoniae* carbapenemase; *n* = 11), OXA-48 (oxacillinase-48; *n* = 14) and VIM (Verona integron-encoded metallo-β-lactamase; *n* = 7). Most of the carbapenemase-producers co-produced ESBL or AmpC β-lactamases, but these are not documented here for the sake of clarity. Of the remaining isolates of Enterobacterales, 47 produced extended spectrum β-lactamases including isolates with CTX-M (*n* = 20), SHV-type (*n* = 19), TEM-type (*n* = 8), and 22 had acquired AmpC β-lactamase including isolates with ACC-1 (*n* = 3), CMY-type (*n* = 6), DHA-1 (*n* = 6), FOX-type (*n* = 3) and LAT-type (*n* = 4). Isolates with acquired AmpC β-lactamase included *K. pneumoniae* (*n* = 11), *P. mirabilis* (*n* = 5), *E. coli* (*n* = 3), *Salmonella* species (*n* = 2) and *K. oxytoca* (*n* = 1). 

A collection of 50 isolates of *Staphylococcus aureus* included 36 strains of MRSA frequently encountered in Europe, including strains isolated in Belgium, Finland, France, Germany, and the United Kingdom. Another MRSA strain, NCTC 11939, was included as a control, as well as a methicillin-susceptible control (NCTC 6571). A further twelve isolates of methicillin susceptible *S. aureus* (MSSA) recently recovered from blood cultures were included. Finally, 50 isolates of enterococci included two control strains (*Enterococcus faecalis* NCTC 775 and *Enterococcus faecium* NCTC 7171) and 48 isolates from clinical samples obtained from at least three different hospitals. The clinical isolates included *E. faecalis* (*n* = 10), *E. faecium* (*n* = 33), *Enterococcus casseliflavus* (*n* = 3), and *Enterococcus gallinarum* (*n* = 2). Of the 50 isolates, 43 were resistant to vancomycin as demonstrated by MIC testing and confirmation of resistance genes by PCR.

### 4.3. Determination of MICs

All MICs were determined using an agar dilution method [21]. This necessitated the use of a defined antagonist-free medium (peptone-free), prepared as previously described [7] with the inclusion of 2% saponin-lysed horse blood, 25 mg/L NAD and 25 mg/L haemin. Test compounds were dissolved in sterile deionised water and incorporated into agar at a concentration range of 0.031–8 mg/L (0.016–32 mg/L for Gram-positive bacteria). All isolates were prepared at a density equivalent to 0.5 McFarland units in sterile deionised water using a densitometer (bioMérieux, Basingstoke, UK) to produce an inoculum of approximately 1.5 × 10^8^ CFU/mL, and then diluted 1 in 15. A 1 µL aliquot of each diluted suspension was then delivered onto plates with a multipoint inoculator (Denley Instruments Limited, Cambridge, UK) to give a final inoculum of 10,000 CFU/spot as recommended [21]. Fosfomycin MICs were performed using the same method except that IsoSensitest agar (Oxoid, Basingstoke, UK) was used supplemented with 25 mg/L glucose-6-phosphate (Sigma, Poole, UK) and an extended range of fosfomycin concentrations. All plates, including the antimicrobial-free control, were incubated for 22 h at 37 °C. 

### 4.4. Study of the Interaction between Alafosfalin and Meropenem Against Carbapenemase-Producing Enterobacterales

Twenty isolates were selected that showed varying susceptibility to meropenem (susceptible, intermediate and resistant) as defined by CLSI guidelines [22]. These included isolates with KPC (*n* = 5), NDM-1 (*n* = 5), OXA-48 (*n* = 5), IMP (*n* = 3) and VIM (*n* = 2) carbapenemases. Interaction was examined using a checkerboard technique with agar plates incorporating various concentrations of meropenem (0, 0.031–64 mg/L) and alafosfalin (0, 0.016–8 mg/L). The medium used was antagonist-free agar (as described above) supplemented with 70 mg/L zinc sulphate to ensure optimal activity of metallo β-lactamases [23]. Plates were inoculated and incubated as described for MIC testing. Fractional inhibitory concentration indices (FICI) were calculated as follows: [(MIC of meropenem in combination with alafosfalin)/(MIC of meropenem alone)] + [(MIC of alafosfalin in combination with meropenem)/(MIC of alafosfalin alone)], and a FICI of ≤0.5 was used to define synergy. All experiments (including MIC testing) were performed on at least two separate occasions to ensure that only reproducible results were reported.

## 5. Concluding Remarks

In light of the emerging threat posed by antimicrobial resistance it is imperative that we find not only new treatments for resistant bacterial infections, but also new diagnostic tools so that the physician can rapidly identify a suitable course of treatment for the presenting patient [24,25]. This paper focuses upon the first of these requirements, and serves as a reminder that there are many useful compounds for treatment of bacterial infections that are already known, but have lay dormant in some cases for decades because at the time of their initial discovery there were many other options available. However, the situation has changed markedly in recent years. We show here that the phosphonopeptides are potentially useful weapons for treating drug resistant infections, including those posed by some of the pathogens that clinicians find to be most challenging at the present time.

As mentioned above it is also important for the clinician to be armed with suitable tools to help distinguish key pathogens so as to be able to determine a suitable course of treatment in a minimal timeframe. One other use of antimicrobial agents is deployment within selective culture media, wherein selectivity in antimicrobial activity can be exploited to suppress the growth of commensal bacteria present within a patient derived sample, thus allowing a non-inhibited pathogen to grow freely which in turn allows for clear and prompt diagnosis [26]. In the period subsequent to the work disclosed within this paper we performed systematic modifications to the structures of the inhibitors with this other goal in mind. The results of these subsequent investigations form the basis of our second and third papers which may be found within this Special Issue of molecules dedicated to microbial detection and identification [27,28]. 

## Figures and Tables

**Figure 1 molecules-25-01445-f001:**
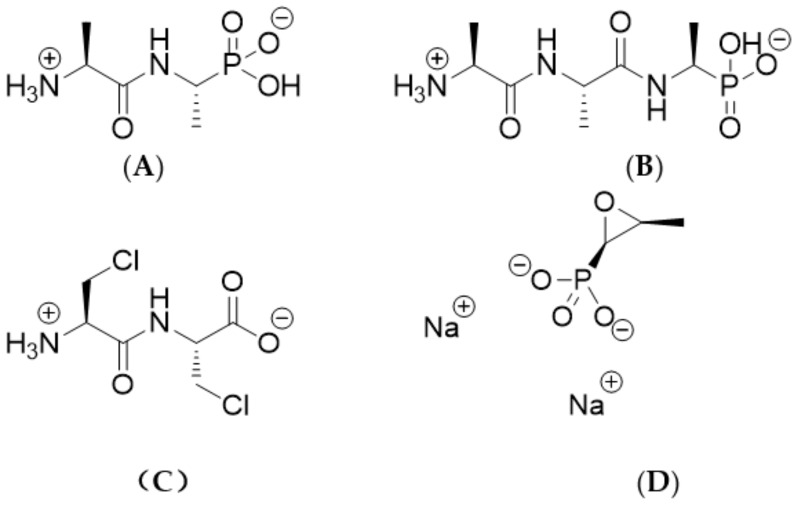
Structures of four compounds used in this study. Systematic names (and abbreviations used in this study) are as follows: (**A**) L-alanyl-L-1-aminoethylphosphonic acid (alafosfalin); (**B**) L-alanyl-L-alanyl-L-1-aminoethylphosphonic acid (di-alanyl fosfalin); (**C**) β-chloro- L-alanyl-β-chloro-L-alanine (β-Cl-Ala-β-Cl-Ala); (**D**) Disodium [(2R,3S)-3-methyloxiran-2-yl]phosphonate (fosfomycin).

**Table 1 molecules-25-01445-t001:** Minimum inhibitory concentrations of various antimicrobial agents against groups of Gram-negative bacteria including isolates with defined resistance mechanisms.

Organism (No. Tested) and Antimicrobial Agent	Concentration (mg/L)			
	Mode	MIC50	MIC90	Range
**Enterobacterales** (*n* = 197)				
Alafosfalin	2	2	> 8	≤0.031–>8
Di-alanyl fosfalin	> 8	8	> 8	≤ 0.031–> 8
β-Cl-Ala-β-Cl-Ala	> 8	> 8	> 8	2–> 8
Fosfomycin	4	4	> 32	0.125–> 32
***E. coli*** (*n* = 53)				
Alafosfalin	0.063	0.125	0.25	≤ 0.031–2
Di-alanyl fosfalin	0.25	0.5	2	≤ 0.031–> 8
β-Cl-Ala-β-Cl-Ala	8	8	> 8	2–> 8
Fosfomycin	0.5	0.5	1	0.125–8
***K. pneumoniae*** (*n* = 87)				
Alafosfalin	2	2	> 8	0.25–> 8
Di-alanyl fosfalin	> 8	> 8	> 8	0.5–> 8
β-Cl-Ala-β-Cl-Ala	> 8	> 8	> 8	8–> 8
Fosfomycin	4	8	> 32	2–> 32
***E. cloacae*** (*n* = 27)				
Alafosfalin	1	1	1	0.125–4
Di-alanyl fosfalin	4	> 8	> 8	0.25–> 8
β-Cl-Ala-β-Cl-Ala	> 8	> 8	> 8	8–> 8
Fosfomycin	16	16	32	4–> 32
**CPE** (*n* = 128)				
Alafosfalin	2	1	4	≤ 0.031–> 8
Di-alanyl fosfalin	> 8	8	> 8	0.063–> 8
β-Cl-Ala-β-Cl-Ala	> 8	> 8	> 8	2–> 8
Fosfomycin	16	4	32	0.125–> 32
**ESBL** (*n* = 47)				
Alafosfalin	2	2	> 8	≤ 0.031–> 8
Di-alanyl fosfalin	> 8	8	> 8	≤ 0.031–> 8
β-Cl-Ala-β-Cl-Ala	> 8	> 8	> 8	2–> 8
Fosfomycin	> 32	8	> 32	0.125–> 32
**AmpC** (*n* = 22)				
Alafosfalin	> 8	4	> 8	0.063–> 8
Di-alanyl fosfalin	> 8	> 8	> 8	0.063–> 8
β-Cl-Ala-β-Cl-Ala	> 8	> 8	> 8	8–> 8
Fosfomycin	32	8	> 32	0.125–> 32

Abbreviations: MIC_50_: concentration of antimicrobial required to inhibit 50% of isolates. MIC_90_: concentration of antimicrobial required to inhibit 90% of isolates. CPE: carbapenemase-producing Enterobacterales; ESBL: Enterobacterales with extended spectrum β-lactamase; AmpC: Enterobacterales with acquired AmpC β-lactamase.

**Table 2 molecules-25-01445-t002:** Minimum inhibitory concentrations of various antimicrobial agents against groups of Gram-positive bacteria including isolates with defined resistance mechanisms.

Organism (No. Tested) and Antimicrobial Agent	Concentration (mg/L)			
	Mode	MIC50	MIC90	Range
**All *S. aureus*** (*n* = 50)				
Alafosfalin	4	4	8	0.125–16
Di-alanyl fosfalin	4	8	16	0.5–32
β-Cl-Ala-β-Cl-Ala	2	2	4	0.125–16
Fosfomycin	8	4	16	0.5–> 32
**MRSA** (*n* = 37)				
Alafosfalin	4	4	8	0.125–16
Di-alanyl fosfalin	4	8	16	0.5–32
β-Cl-Ala-β-Cl-Ala	2	2	4	0.125–16
Fosfomycin	8	4	16	0.5–> 32
**MSSA** (*n* = 13)				
Alafosfalin	4	4	8	0.25–16
Di-alanyl fosfalin	16	8	16	0.5–32
β-Cl-Ala-β-Cl-Ala	1	1	2	0.5–2
Fosfomycin	4	4	16	2–16
**All Enterococci** (*n* = 50)				
Alafosfalin	8	16	> 32	4–> 32
Di-alanyl fosfalin	0.5	0.5	2	≤ 0.016–> 32
β-Cl-Ala-β-Cl-Ala	16	16	32	2–16
Fosfomycin	> 32	> 32	> 32	16–> 32
***E. faecalis*** (*n* = 11)				
Alafosfalin	8	8	32	4–> 32
Di-alanyl fosfalin	0.031	0.063	0.5	≤ 0.016–> 32
β-Cl-Ala-β-Cl-Ala	8	8	16	4–16
Fosfomycin	32	32	> 32	32–> 32
***E. faecium*** (*n* = 34)				
Alafosfalin	16	16	16	4–32
Di-alanyl fosfalin	0.5	0.5	2	≤ 0.016–4
β-Cl-Ala-β-Cl-Ala	16	16	32	2–> 32
Fosfomycin	> 32	> 32	> 32	16–> 32
**GRE** (*n* = 43)				
Alafosfalin	16	16	> 32	4–> 32
Di-alanyl fosfalin	0.5	0.5	> 32	≤ 0.016–> 32
β-Cl-Ala-β-Cl-Ala	16	16	> 32	4–> 32
Fosfomycin	> 32	> 32	> 32	32–> 32

Abbreviations: MIC_50_: concentration of antimicrobial required to inhibit 50% of isolates. MIC_90_: concentration of antimicrobial required to inhibit 90% of isolates. MRSA: methicillin-resistant *S. aureus*; MSSA: methicillin-susceptible *S. aureus*; GRE: glycopeptide-resistant enterococci.

**Table 3 molecules-25-01445-t003:** Interaction between alafosfalin and meropenem against carbapenemase-producing Enterobacterales as determined using a chequerboard technique.

Species	Carbapenemase	Alafosfalin MIC (mg/L)	Meropenem MIC (mg/L)	FICI	Interpretation
*K. oxytoca*	KPC-2	4	0.25	0.16	Synergy
*K. pneumoniae*	KPC-3	1	1	0.25	Synergy
*K. pneumoniae*	KPC-3	1	16	0.19	Synergy
*K. pneumoniae*	KPC-4	0.5	4	0.38	Synergy
*K. pneumoniae*	KPC	1	0.25	0.38	Synergy
*E. coli*	NDM-1	0.125	8	0.75	No interaction
*E. coli*	NDM-1	0.031	0.125	0.76	No interaction
*E. cloacae*	NDM-1	1	4	1	No interaction
*K. pneumoniae*	NDM-1	2	2	0.27	Synergy
*E. coli*	NDM-1	0.125	16	0.25	Synergy
*E. cloacae*	VIM-1	2	4	0.19	Synergy
*K. pneumoniae*	VIM-1	2	16	0.28	Synergy
*K. pneumoniae*	IMP	0.25	1	0.37	Synergy
*K. pneumoniae*	IMP	0.25	1	0.50	Synergy
*K. oxytoca*	IMP	1	0.5	0.13	Synergy
*K. pneumoniae*	OXA-48	1	8	0.38	Synergy
*E. coli*	OXA-48	0.125	0.25	0.37	Synergy
*K. pneumoniae*	OXA-48	2	1	0.5	Synergy
*K. pneumoniae*	OXA-48	2	4	0.26	Synergy
*K. pneumoniae*	OXA-48	1	2	0.63	No interaction

Abbreviation: FICI; Fractional inhibitory concentration index.

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
