# Peer review of "Phosphonopeptides Revisited, in an Era of Increasing Antimicrobial Resistance"

_molecules, 2020, doi:10.3390/molecules25061445_

Round 1
Reviewer 1 Report
The paper “Phosphonopeptides revisited, in an era of increasing antimicrobial resistance” evaluates the antibacterial activity of three different peptide mimetics (alafosfalin, di-alanyl fosfalin and ß-chloro-L-alanyl-ß-chloro-L-alanine with a collection of 297 bacteria including many multi-drug resistant strains.
The paper falls into the aims of the journal. The study design is clear, and the methodology used is adequate for the purposes. The findings are consistent with current knowledge of the literature.
There are no significant criticisms about this paper.
As a minor points, I think that it should be better clarified that the compound ß-chloro-L-alanyl-ß-chloro-L-alanine seems to have a weak or null activity on Gram-negative isolates, as well as the di-alanyl fosfalin, and that overall the activity of the ß-chloro-L-alanyl-ß-chloro-L-alanine is relative only to staphylococci, whereas the di-alanyl fosfalin could be developed against Gram positive microorganisms in general.
The paper evaluates the MICs of Fosfomycin by using the broth microdilution technique, which is not considered the gold standard for this drug (the agar dilution technique is preferred): this should be discussed as a limitation of the study.
It is quite challenging to understand the discrepancy between the data obtained for E. cloacae, demonstrating an excellent performance of alafosfalin, and those obtained for the “AmpC” group (alafosfalin MICs mode >8 mcg/ml). AmpC is constitutive for Enterobacter cloacae, and it is, therefore, unclear which microorganisms are included in the “AmpC” group and the differences in the MIC modes. This should be better explained in the text.
In the discussion, it could be discussed the possible clinical use of alafosfalin based on the pK/pD parameters: I agree that this compound shows a good activity on microorganisms which can be difficult to treat, but is not clear from the data presented if the achievable serum concentrations can be significantly higher than MIC modes (line 122, 500 mg allow to reach a concentration equal to the MIC mode) and if the short half-life (more or less, one hour) can impact on the drug usage
Author Response
As a minor points, I think that it should be better clarified that the compound ß-chloro-L-alanyl-ß-chloro-L-alanine seems to have a weak or null activity on Gram-negative isolates, as well as the di-alanyl fosfalin, and that overall the activity of the ß-chloro-L-alanyl-ß-chloro-L-alanine is relative only to staphylococci, whereas the di-alanyl fosfalin could be developed against Gram positive microorganisms in general.
Author response: Thank you for the review of our paper. It is certainly the case that ß-chloro-L-alanyl-ß-chloro-L-alanine has relatively poor activity against Gram-negative bacteria. We have added text to line 92 to state that: “Compared to alafosfalin, ß-Cl-Ala-ß-Cl-Ala showed relatively poor activity against Gram-negative bacteria but it showed the highest activity against S. aureus”.
We would suggest that “weak activity” against Gram-negative bacteria by di-alanyl fosfalin is less clear-cut. For example, 90% of E. coli isolates were inhibited by 2 mg/L of di-alanyl fosfalin. It is true however that its activity against other Gram-negative species is limited. We had previously stated that: “As we have shown in this study, other peptide mimetics including di-alanyl fosfalin have substantive in vitro activity against multi-drug resistant bacteria”. This has been amended to state (more specifically) that “As we have shown in this study, other peptide mimetics, such as di-alanyl fosfalin, may show substantial antimicrobial activity against E. coli and particularly enterococci.” (Line 194).
The paper evaluates the MICs of Fosfomycin by using the broth microdilution technique, which is not considered the gold standard for this drug (the agar dilution technique is preferred): this should be discussed as a limitation of the study.
Author response: As originally stated in line 201 (now line 235): “All MICs were determined using an agar dilution method”. We agree with the referee that this is the correct method for determination of fosfomycin MICs. The broth microdilution technique was not used.
It is quite challenging to understand the discrepancy between the data obtained for E. cloacae, demonstrating an excellent performance of alafosfalin, and those obtained for the “AmpC” group (alafosfalin MICs mode >8 mcg/ml). AmpC is constitutive for Enterobacter cloacae, and it is, therefore, unclear which microorganisms are included in the “AmpC” group and the differences in the MIC modes. This should be better explained in the text.
Author response: Agreed. This is a very useful and astute observation that will improve the paper. The reason for this apparent discrepancy is due to the composition of species in the AmpC group, which was undisclosed in the first version of this paper. The data are particularly skewed by the inclusion of Proteus mirabilis (n = 5) and Salmonella species (n = 2) – as these are species which have relatively high levels of intrinsic resistance to alafosfalin. The species that comprise the “AmpC group” have now been disclosed in line 221. The reason for this apparent discrepancy should now be much clearer to readers.
In the discussion, it could be discussed the possible clinical use of alafosfalin based on the pK/pD parameters: I agree that this compound shows a good activity on microorganisms which can be difficult to treat, but is not clear from the data presented if the achievable serum concentrations can be significantly higher than MIC modes (line 122, 500 mg allow to reach a concentration equal to the MIC mode) and if the short half-life (more or less, one hour) can impact on the drug usage
Author response:
Although this is clearly a very important point, these are complex questions that we feel are beyond the scope of our in vitro study and, frankly, a little outside of our area of expertise. We have cited at least 5 papers that provide insights into PK/PD parameters including papers that demonstrate successful outcomes in a mouse septicaemia model and concentration of alafosfalin in urine to levels that are well in excess of modal MICs. We do not feel comfortable adding to the existing text and we wish to stringently avoid over stating the clinical utility of alafosfalin when it seems clear that much more work is required in this area. We feel that the discussion, as presented, provides sufficient evidence to show the clinical potential of alafosfalin, rather than demonstrating proof of clinical utility.
Reviewer 2 Report
In general, the manuscript since my point of view is adequate. However, two things need to correct.
First: The author confound the word “Enterobacterales” with Enterobacteriaceae. The former word refer to Enterobacter genera group and not for the enterobacteria family. The second one is better to use that include all members of enterobacteria family (Enterobacteriaceae).
Second: Even the people that are working with different resistance enzymes (carbapenemase), the author must include the meaning of all abbreviation in the manuscript or references related to such abbreviations.
Author Response
In general, the manuscript since my point of view is adequate. However, two things need to correct.
First: The author confound the word “Enterobacterales” with Enterobacteriaceae. The former word refer to Enterobacter genera group and not for the enterobacteria family. The second one is better to use that include all members of enterobacteria family (Enterobacteriaceae).
Author response: Thank you for the review of our paper. We defend the use of the term Enterobacterales which encompasses all of the Gram-negative species documented here (rather than just Enterobacter, as implied). We take our lead from the taxonomic changes published relatively recently by Adeolu et al. (Int J Syst Evol Microbiol. 2016 Dec;66(12):5575-5599) and the adoption of this nomenclature by international organisations such as EUCAST (e.g. here: http://www. eucast.org/ clinicalbreakpoints/) .
Second: Even the people that are working with different resistance enzymes (carbapenemase), the author must include the meaning of all abbreviation in the manuscript or references related to such abbreviations.
Author response: The abbreviations used for the five different types of carbapenemase have now been added to the paper in line 214.
Reviewer 3 Report
Perry and co-workers present a kind of drug repurposing work in which three peptide mimetics (e.g. alafosfalin) were resynthesized and their activity were studied against 297 bacterial isolates including glycopeptide resistant enterococci, methicillin resistant staphylococci and carbapenemase-producing Enterobcterales (CPE). The synergestic effect of alafosfalin and meropenem against 20 CPE isolates was also investigated.
Biological evaluation of dormant (forgotten) antimicrobial agents is a fashionable and promising strategy for combating multidrug-resistant bacteria. The present manuscript reports an important and valuable work in this field, demonstrating that phosphonopeptides, e.g. alaphosphaline
may be used for treating drug resistant infection.
The biological results are clearly presented and the synthesis of the compounds in the SI is properly described.
I recommend this article for publication in Moecules after minor revision.
My comments:
- The newest reference cited in the Introduction is from 2012. It would be useful to cite some new literatures on antibiotic resistance.
- It is stated in the manuscript (page 2) that the results of synergy testing are shown in Table 2. However, there is no Table 2 in the manuscript! Without Table 2, the manuscript the work cannot be published!
Author Response
Perry and co-workers present a kind of drug repurposing work in which three peptide mimetics (e.g. alafosfalin) were resynthesized and their activity were studied against 297 bacterial isolates including glycopeptide resistant enterococci, methicillin resistant staphylococci and carbapenemase-producing Enterobcterales (CPE). The synergestic effect of alafosfalin and meropenem against 20 CPE isolates was also investigated.
Biological evaluation of dormant (forgotten) antimicrobial agents is a fashionable and promising strategy for combating multidrug-resistant bacteria. The present manuscript reports an important and valuable work in this field, demonstrating that phosphonopeptides, e.g. alaphosphaline may be used for treating drug resistant infection.
The biological results are clearly presented and the synthesis of the compounds in the SI is properly described.
I recommend this article for publication in Molecules after minor revision.
My comments:
1. The newest reference cited in the Introduction is from 2012. It would be useful to cite some new literatures on antibiotic resistance.
Author response: Thank you for the review of our paper. In 2012, a number of seminal papers were published that summarized the crisis in antibiotic development. Despite having reviewed more recent papers, we believe that these particular papers are the most useful to emphasise the points we make in the introduction. However, we have updated reference 5 in the introduction to refer to a new paper from 2020. We have also added two more recent papers that we think are best referenced in our concluding remarks (line 265).
2.It is stated in the manuscript (page 2) that the results of synergy testing are shown in Table 2. However, there is no Table 2 in the manuscript! Without Table 2, the manuscript the work cannot be published!
Author response: We apologize for this careless omission. The table has now been included (now labelled as Table 3). As Table 1 was very large (and unlikely to fit on a single page), it has been divided into two: Table 1 for Gram-negative bacteria and Table 2 for Gram-positive bacteria.